# Structural basis of cyclobutane pyrimidine dimer recognition by UV-DDB in the nucleosome

Syota Matsumoto [1], Yoshimasa Takizawa [1,2], Mitsuo Ogasawara[1], Kana Hashimoto[1], Lumi Negishi [1], Wenjie Xu[1,3], Haruna Tachibana[1,3], Junpei Yamamoto [4], Shigenori Iwai[4], Kaoru Sugasawa [5] & Hitoshi Kurumizaka [1,3,6] ✉

In mammalian global genomic nucleotide excision repair, UV-DDB plays a central role in recognizing DNA lesions, such as 6-4 photoproducts and cyclobutane pyrimidine dimers, within chromatin. In the present study, we perform cryo-electron microscopy analyses coupled with chromatin-immunoprecipitation to reveal that the cellular UV-DDB binds to UV-damaged DNA lesions in a chromatin unit, the nucleosome, at a position approximately 20 base-pairs from the nucleosomal dyad in human cells. An alternative analysis of the in vitro reconstituted UV-DDB-cyclobutane pyrimidine dimer nucleosome structure demonstrates that the DDB2 subunit of UV-DDB specifically recognizes the cyclobutane pyrimidine dimer lesion at this position on the nucleosome. We also determine the structures of UV-DDB bound to DNA lesions at other positions in purified cellular human nucleosomes. These cellular and reconstituted UV-DDB-nucleosome complex structures provide important evidence for understanding the mechanism by which UV lesions in chromatin are recognized and repaired in mammalian cells.

In the eukaryotic genome, 145-147 base-pair DNA segments are wrapped around histone octamers, consisting of two molecules each of the four types of histone proteins, H2A, H2B, H3, and H4, forming the basic unit called the nucleosome[1,2]. The chromatin structure consists of sequential nucleosomes and regulates various nuclear events, including the DNA damage response[3]. Chromatin is considered to be a barrier for proteins to gain access to DNA. However, it is unclear how DNA binding proteins overcome this limitation for their activity[4].

The DNA damage response is an important mechanism to maintain genome integrity across generations. DNA damage generates chemical conformational changes, forming obstacles that block transcription and induce mutations to the genome during DNA replication.

Ultraviolet (UV) light causes DNA damage and generates mainly two types of DNA lesions, the (6-4)photoproduct (6-4PP) and the cyclobutane pyrimidine dimer (CPD)[5].

Nucleotide excision repair (NER) is an important cellular system that eliminates a variety of DNA lesions. Defects in NER cause severe diseases, such as xeroderma pigmentosum (XP) and Cockayne syndrome (CS), which are characterized by sensitivity to sunlight, and XP is associated with a high frequency of skin cancer[6,7]. NER comprises two major pathways: global genomic NER (GG-NER) and transcription-coupled NER (TC-NER). In mammalian GG-NER, it is important for XPC to recognize DNA lesions. Although XPC binds to 6-4PP efficiently and proceeds NER, the binding affinity of XPC against CPD is shown to be

[1]Laboratory of Chromatin Structure and Function, Institute for Quantitative Biosciences, The University of Tokyo, Bunkyo-ku, Tokyo, Japan. [2]Department of Computational Biology and Medical Sciences, Graduate School of Frontier Sciences, The University of Tokyo, Bunkyo-ku, Tokyo, Japan. [3]Department of Biological Sciences, Graduate School of Science, The University of Tokyo, Bunkyo-ku, Tokyo, Japan. [4]Division of Chemistry, Graduate School of Engineering Science, Osaka University, Toyonaka-shi, Osaka, Japan. [5]Biosignal Research Center, Organization of Advanced Science and Technology, Kobe University, Kobe-shi, Hyogo, Japan. [6]RIKEN Center for Biosystems Dynamics Research, Yokohama, Japan. ✉e-mail: kurumizaka@iqb.u-tokyo.ac.jp

quite low in vitro[8]. Therefore, UV-DDB (DDB1-DDB2/XPE heterodimer complex) is thought to be responsible for recognizing CPD in the genome, and then handover the lesion to XPC. In this process, while XPC is monoubiquitinated, the DDB2 subunit is polyubiquitinated, dissociated from the damaged site by the p97/VCP segregase, and subsequently degraded by the 26S proteasome[9–13]. The XPC-RAD23B-centrin2 (XPC complex) bound to the DNA lesion then recruits other downstream factors to complete the entire NER process[14,15]. However, the access of NER factors to DNA lesions is limited by the chromatin structures in cells[16,17].

The mechanism by which UV-DDB gains access to DNA damage has been studied in vitro. DDB2/XPE is the DNA-binding subunit of UV-DDB and has high affinity to the major DNA lesions, 6-4PP and CPD[18]. In addition to bulky lesions, DDB2 can also bind to abasic sites in DNA[18]. UV-DDB reportedly functions as a damage sensor in the base excision repair (BER) pathway in chromatin[19–21]. Previous X-ray crystallographic analyses revealed the structures of UV-DDB-damaged DNA complexes, in which DDB2 binds to UV lesions using its positively charged surface[22–24]. The β-hairpin loop of DDB2 is responsible for stabilizing the interactions between the amino acid residues (F-Q-H) and the two orphan bases opposite a UV DNA lesion site. DDB2 binding to UV DNA lesions mainly occurs when the lesion is induced on the strand facing toward the solvent, and this orientation preference is also conserved in the recognition of nucleosomal 6-4PP and abasic sites[25,26]. In contrast, to repair the 6-4PP and abasic DNA lesions facing toward the histone surface in the nucleosome, DDB2 binding induces register shifting of the nucleosomal DNA to expose the DNA lesion toward the accessible surface[26]. However, the mechanism by which the UV-DDB complex recognizes CPD in nucleosomes remains enigmatic, although it is the major lesion caused by UV light.

To gain insight into UV-DDB binding to UV DNA lesions in cells, in the present study, we performed chromatin-immunoprecipitation coupled cryo-electron microscopy (ChIP-CryoEM)[27], and visualized UV-DDB binding to damaged nucleosomal DNA in cells. We also performed cryo-electron microscopy (cryo-EM) analyses with reconstituted and native human nucleosomes containing UV DNA lesions, as complexes with purified UV-DDB. These native and reconstituted structures revealed the mechanism by which UV-DDB recognizes the UV lesion in chromatin.

## Results

### Preparation of cellular UV-DDB bound to nucleosomes in human cells
To determine how UV-DDB recognizes DNA damage in cellular chromatin, we established a cell line (U2OS/FLAG-DDB2) in which FLAG-tagged DDB2, the DNA-binding subunit of UV-DDB, is stably produced in the DDB2-deficient cells (U2OS/DDB2 KO)[28]. To test the localization of FLAG-DDB2, we irradiated UV-C, which is reported to generate mainly CPD within the genome. Immunostaining showed that without UV damage, FLAG-DDB2 was evenly distributed in the nucleus (Fig. 1a, upper panels). In contrast, upon local UV irradiation, FLAG-DDB2 was co-localized with CPD damage sites (Fig. 1a, lower panels). These results suggested that, in cells, FLAG-DDB2 recognizes UV damage and binds to UV lesions in chromatin. FLAG-DDB2 was accumulated in chromatin (insoluble fraction) just after UV irradiation (0.5 h) and released after 1 h (Supplementary Fig. 1a). Therefore, UV-DDB may transiently associate with chromatin in a UV damage-dependent manner.

A lysate was prepared from U2OS/FLAG-DDB2 cells after UV exposure. The resulting chromatin sample was crosslinked with formaldehyde and digested by sonication and micrococcal nuclease treatment. The cellular UV-DDB bound to chromatin was then immunopurified by agarose beads conjugated with the anti-FLAG M2 antibody (Fig. 1b). After UV-DDB immunopurification, we detected DDB1 and FLAG-DDB2 together with the four types of histones (Fig. 1c).

These proteins were observed by the same chromatin isolation method without crosslinking and confirmed by CBB staining and a mass spectrometry analysis (Supplementary Fig. 1b, c). Our enzyme-linked immunosorbent assay (ELISA) and DNA purification assay confirmed that the eluted fraction contains about 100-150 base-pair DNA fragments bearing CPD lesions (Fig. 1d, Supplementary Fig. 1d).

It should be noted that our mass spectrometric analysis revealed that the pull-down fraction with FLAG-DDB2 contained the peptides from CUL4A, CUL4B, and RBX1, which belong to the ubiquitin-proteasome pathway (Supplementary Fig. 1c). However, their target, ubiquitinated DDB2 (Ub-DDB2), was not observed in the fraction. We surmised that the efficient ubiquitination of DDB2 may decrease its chromatin binding affinity, leading to the dissociation of Ub-DDB2 from the nucleosome.

### ChIP-CryoEM analysis of UV-DDB bound to nucleosomes in human cells
We conducted a ChIP-CryoEM single particle analysis with the immunopurified UV-DDB-nucleosome complexes[27], and visualized the structure of UV-DDB bound to the cellular nucleosome (Fig. 2a, Supplementary Fig. 2a–f, Supplementary Table 1). The cryo-EM maps of the DDB1 and DDB2 subunits and the nucleosome were clearly distinguished. Since only 7339 particles of the UV-DDB-nucleosome complex were obtained, its resolution was limited to 12 Å, which precluded the identification of the amino acid residues (12 Å) (Fig. 2a). However, the 12 Å resolution could reveal the approximate binding position of UV-DDB on the nucleosomal DNA. We could not observe the cryo-EM density for CUL4-RBX1 in the ChIP-CryoEM samples, probably due to the flexible nature of the CUL4-ubiquitin ligase in the complex[23].

In this complex, UV-DDB gained access to the nucleosomal DNA. Superimposition of a high-resolution 3D structure model of the nucleosome (PDB ID: 3AFA) on the cryo-EM map corresponding to the nucleosome suggested that the UV-DDB may bind to the superhelical location (SHL) ± 2 position, which is approximately 20 base-pairs away from the nucleosomal dyad (SHL 0) (Fig. 2b, Supplementary Fig. 3a, b). The DDB1 and DDB2 subunits fit well in the cryo-EM map of the UV-DDB-nucleosome complex (Supplementary Fig. 3c, d). This structure of the cellular UV-DDB-nucleosome complex suggested that the UV-induced DNA damage at the SHL ± 2 position can be efficiently recognized by UV-DDB in cells.

We next tested whether the UV-DDB binding to the SHL ± 2 position is a consequence of its coupling with the ATP hydrolysis-dependent nucleosome remodeling activity or its preferential interaction with the SHL ± 2 position. To do so, we repeated the ChIP-CryoEM analysis in the presence of ATPγS, which is a slowly hydrolyzable analog of ATP (often referred to as a non-hydrolyzable analog). Our pull-down assay revealed that, in the presence of ATPγS, only minor amounts of UV-DDB and histones were captured by agarose beads conjugated with the anti-FLAG M2 antibody (Supplementary Fig. 1e). In addition, the immunopurified UV-DDB-nucleosome complex was not observed in the presence of ATPγS (Supplementary Fig. 1e, f). These results suggest that the preferential UV-DDB binding to the nucleosomal SHL ± 2 position may be induced by its coupling with the ATP hydrolysis-dependent nucleosome remodeling.

### UV-DDB bound to CPD lesions in the nucleosome at SHL ± 2
The UV-C-generated CPD is a major UV lesion in nucleosomes[16,29,30], suggesting that the cellular UV-DDB-nucleosome complex observed by the ChIP-CryoEM method may contain the CPD as UV-induced DNA damage. However, the structure of UV-DDB bound to a nucleosome containing a CPD has not been reported so far. We prepared the nucleosome core particle (NCP) containing a CPD at a position 22-23 base-pairs from the nucleosomal dyad, corresponding to the SHL−2 position (NCP$^{CPD}$) (Fig. 3a, Supplementary Fig. 3e, f). The

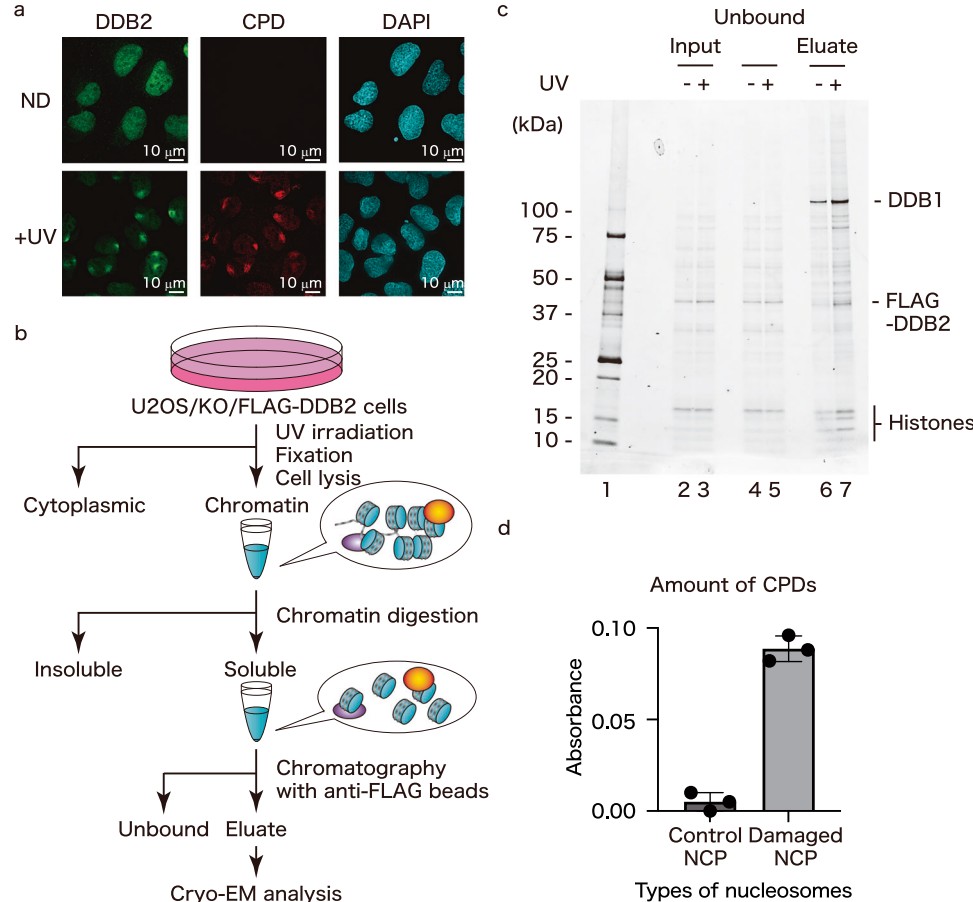

**Fig. 1 | ChIP-CryoEM technique for the cellular UV-DDB-NCP complex. a** U2OS/KO/FLAG-DDB2 cells were immunostained by an anti-DDB2 antibody (green) or an anti-CPD antibody (red). DNA was stained by DAPI (cyan). **b** Workflow of the ChIP-CryoEM method. **c** Soluble (Input: lanes 2,3) after MNase treatment, Unbound (Unb.: lanes 4,5), and bound (Eluate: lanes 6,7) fractions after an incubation with anti-FLAG beads were subjected to SDS-PAGE and stained with Oriole (Bio-Rad). Non-treated cells or UV-irradiated cells are indicated as UV- (lanes 2,4,6) or UV+ (lanes 3,5,7). **d** Amounts of CPDs were measured by ELISA, using an anti-CPD antibody. Non-treated cells and UV-irradiated cells were used as control NCPs and damaged NCPs, respectively. The Y-axis represents the absorbance at 490 nm, normalized to the minimum value across all measurements, which was set to zero. The graph was generated using Prism 10 software, and all data are based on three technical replicates and are presented as mean ± s.d.

UV-DDB-NCP$^{CPD}$ complex was reconstituted in vitro and subjected to the cryo-EM single particle analysis. We then determined the cryo-EM structure of the UV-DDB-NCP$^{CPD}$ complex at 3.4 Å resolution (Fig. 3b, Supplementary Fig. 4a–f, Supplementary Fig. 5a–h, Supplementary Table 1). Interestingly, the UV-DDB-NCP$^{CPD}$ structure fits very well into the cryo-EM map of the cellular UV-DDB-NCP complex (Fig. 3c). Therefore, the cellular UV-DDB-NCP complex obtained from human cells may be a form in which UV-DDB binds to the UV-induced DNA lesion at the SHL ± 2 position.

In the UV-DDB-NCP$^{CPD}$ structure, the F334, Q335, and H336 residues of DDB2 bound to the two orphan bases opposite the lesion (Fig. 4a, b, Supplementary Fig. 5e). The DDB2 G154, A155, G156, I200, N201, and W203 residues directly contacted the CPD moiety of the nucleosomal DNA. Especially, the DDB2 I200 and N201 residues recognized the two methyl groups of the CPD (Fig. 4c, Supplementary Fig. 5f). In contrast, the I200 and N201 residues that contacted the CPD in the UV-DDB-NCP$^{CPD}$ structure are located away from the lesions in the UV-DDB complexes with nucleosomes containing the 6-4PP or abasic site (Supplementary Fig. 5f, g)[26]. The positively charged surface of DDB2, comprising the R112, K132, K244, and Q308 residues, bound to the DNA backbone near the CPD site. Among these residues, the DDB2 K244E mutation has been found in XP-E patients[31]. The DDB2 K244E mutant is reportedly unable to bind to the CPD[32]. This fact suggested that the nucleosomal DNA binding near the CPD lesion by the positively charged DDB2 surface plays an important role in the NER pathway. In addition, the DDB2 R332, Y356, I394, R370, and Y413 residues interacted with the backbone phosphates of the undamaged strand within the CPD site (Supplementary Fig. 5h, i). In the CPD nucleosome complexed with UV-DDB, the bending angle of the nucleosomal DNA at the CPD site was 53°, while it was 49° for undamaged nucleosomal DNA (Fig. 4d, e, Supplementary Fig. 5j). In contrast, the bending angle of non-nucleosomal DNA was reported to be ~45° at the CPD site upon DDB2 binding[23]. Therefore, UV-DDB induces DNA bending at the CPD site, even when the CPD site is already bent by the histones in the nucleosome.

## UV-DDB binds to multiple damaged DNA sites in the nucleosome

We next tested whether UV-DDB binds to other damaged DNA positions in the nucleosome. To do so, we prepared native nucleosomes from Expi293 cells, which produce FLAG-tagged H2A (Fig. 5a). The cells were lysed, and the chromatin was digested by an MNase treatment. The resulting mononucleosomes were isolated by immunopurification using agarose beads conjugated with the anti-FLAG M2 antibody, and UV irradiated in vitro (UV-irradiated NCP) (Fig. 5a). We confirmed that the nucleosome prepared from the H2A-FLAG producing Expi293 cells contains histones H2A-FLAG, H2A, H2B, H3, and H4, as expected, and the histone composition was not affected by UV irradiation (Fig. 5b). Since the purified native nucleosomes contain various genomic DNA

sequences, the UV damage is potentially introduced at various nucleosome positions.

We then tested UV-DDB binding to UV-damaged nucleosomes with native genomic DNA sequences. UV-DDB specifically bound to the UV-damaged nucleosomes and formed two distinct complexes, probably containing one and two UV-DDB complexes (Fig. 5c, lanes 6-10). These complexes were scarcely detected with the native nucleosomes without UV damage (Fig. 5c, lanes 1-5). Consistently, the UV-irradiated nucleosomes were confirmed to contain CPDs, although 6-4PP lesions may also have been present (Fig. 5d).

Our cryo-EM single-particle analysis revealed that UV-DDB bound to multiple sites in the nucleosome, primarily at three nucleosomal DNA positions, SHL ± 2, SHL ± 3, and SHL ± 6. For improved resolution, the DDB1 subunit was masked out, enabling visualization of the DDB2 subunit in the nucleosomes (Fig. 6a, Supplementary Fig. 6, Supplementary Fig. 7a–e, Supplementary Table 1). Superimposition of the UV-DDB and nucleosome structures in these SHL ± 2, SHL ± 3, and

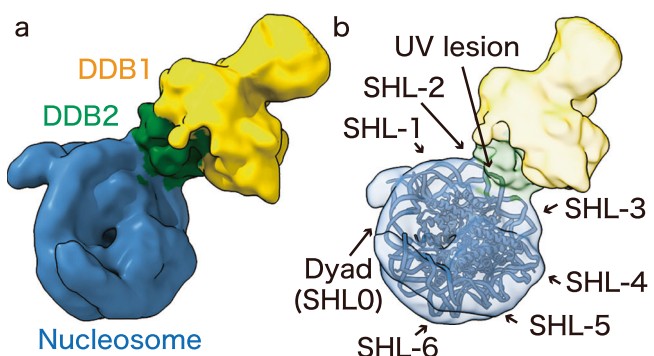

**Fig. 2 | CryoEM map of cellular UV-DDB-NCP complex. a** Cryo-EM map of cellular UV-DDB bound to nucleosomes from cells. The nucleosome, DDB1, and DDB2 are colored blue, yellow, and green, respectively. **b** The 3D structure model of the nucleosome (blue)(PDB ID: 3AFA) is superimposed on the cryo-EM maps. The dyad and each super helical location (SHL) are indicated. The binding site of DDB2 was predicted to be the SHL ± 2 position.

SHL ± 6 complexes suggested that the UV-DDB may bind to the positions 22-23 base-pairs (SHL ± 2), 32-33 base-pairs (SHL ± 3), and 63-64 base-pairs (SHL ± 6) from the nucleosomal dyad. The SHL ± 2 complex is similar to the cellular UV-DDB-nucleosome complex observed by the ChIP-CryoEM analysis (Fig. 2a, b). In the SHL ± 3 and SHL ± 6 complexes, the UV lesions may be exposed to the solvent for efficient recognition by UV-DDB (Fig. 6b). These results suggested that UV-DDB has the potential to recognize UV-induced DNA lesions at multiple positions in the nucleosome. The SHL ± 2 position of the nucleosome is frequently served as the binding site for the nucleosome-binding proteins, such as chromatin remodelers[33,34]. The DNA lesion may relocate to the SHL ± 2 position, which is more accessible for the nucleosome-binding proteins, including UV-DDB. In this process, the DNA register shifting activity of UV-DDB may also contribute to lesion relocation[26].

## Discussion

UV-DDB functions in the initial recognition step of UV-damaged DNA in cells[10]. A previous structural study reported that purified UV-DDB binds to the 6-4PP or abasic damage at the SHL ± 2 position of the reconstituted nucleosome[26]. In the present study, our ChIP-CryoEM technique revealed the structure of the cellular UV-DDB-nucleosome complex in human cells (Fig. 2a, b, Fig. 3c). Interestingly, in this complex, UV-DDB binds to the SHL ± 2 position of the nucleosome. We then reconstituted the UV-DDB-NCP^CPD complex, in which the CPD lesion is exposed to the solvent at the SHL ± 2 position in the reconstituted nucleosome (Fig. 3b). The cryo-EM map of this cellular UV-DDB-nucleosome complex fits very well with that of the reconstituted UV-DDB-NCP^CPD complex. These results indicate that UV-DDB binds to the UV-damaged nucleosome at the SHL ± 2 position in human cells. Importantly, our cryo-EM analysis of UV-DDB binding to the cellular UV-damaged nucleosomes revealed that UV-DDB also binds to the nucleosomal SHL ± 3 and SHL ± 6 positions, in addition to the SHL ± 2 position. In these SHL ± 3 and SHL ± 6 complexes, UV-DDB may bind to exposed UV-damaged sites at the SHL ± 3 and SHL ± 6 positions. Therefore, UV-DDB may recognize UV-induced DNA damage in the nucleosome when the damaged site is located on an accessible surface,

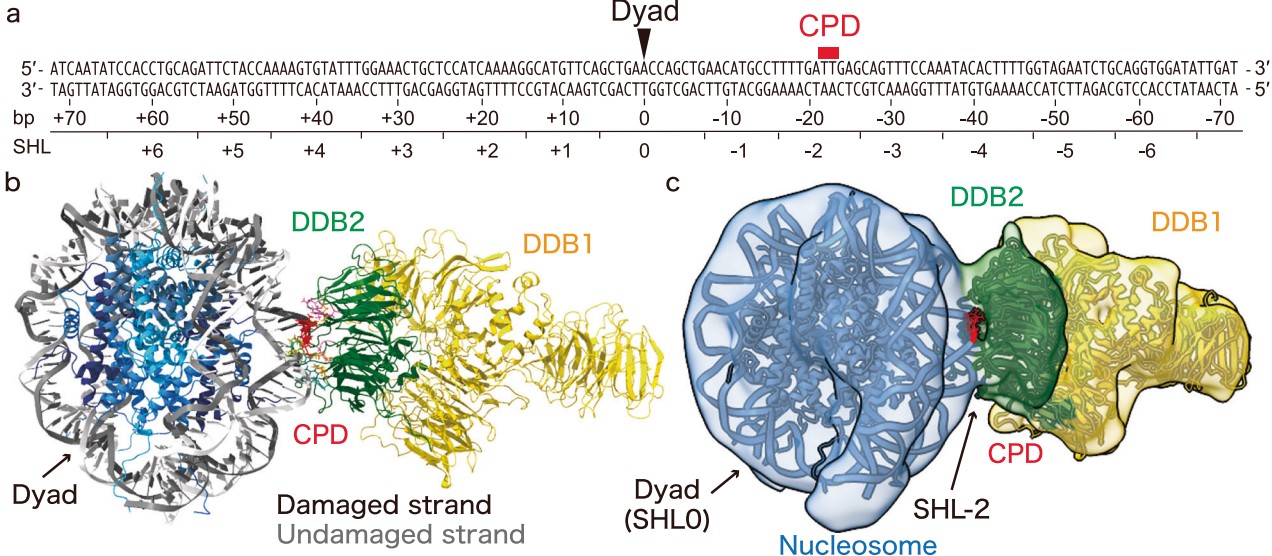

**Fig. 3 | Cryo-EM structure of UV-DDB-NCP^CPD complex. a** DNA sequence with the CPD placed at the position 22-23 base-pairs from the nucleosomal dyad, corresponding to the SHL ± 2 position. **b** The structural model of the UV-DDB-NCP^CPD complex (PDB ID: 9J8W) was determined, corresponding to the cryo-EM map of UV-DDB-NCP^CPD complex at 3.4 Å resolution. DDB1 (yellow), DDB2 (green), CPD (red), damaged DNA strand (light gray), and undamaged DNA strand (dark gray) are highlighted. DDB1 (PDB ID: 6R8Y) was superimposed on the cryo-EM map filtered by a 5 Å low-pass filter. **c** Structural models of the UV-DDB-NCP^CPD architecture (PDB ID: 9J8W) and DDB1 (PDB ID: 6R8Y) were superimposed on cryo-EM maps of the cellular UV-DDB-NCP complex shown in Fig. 2a, b.

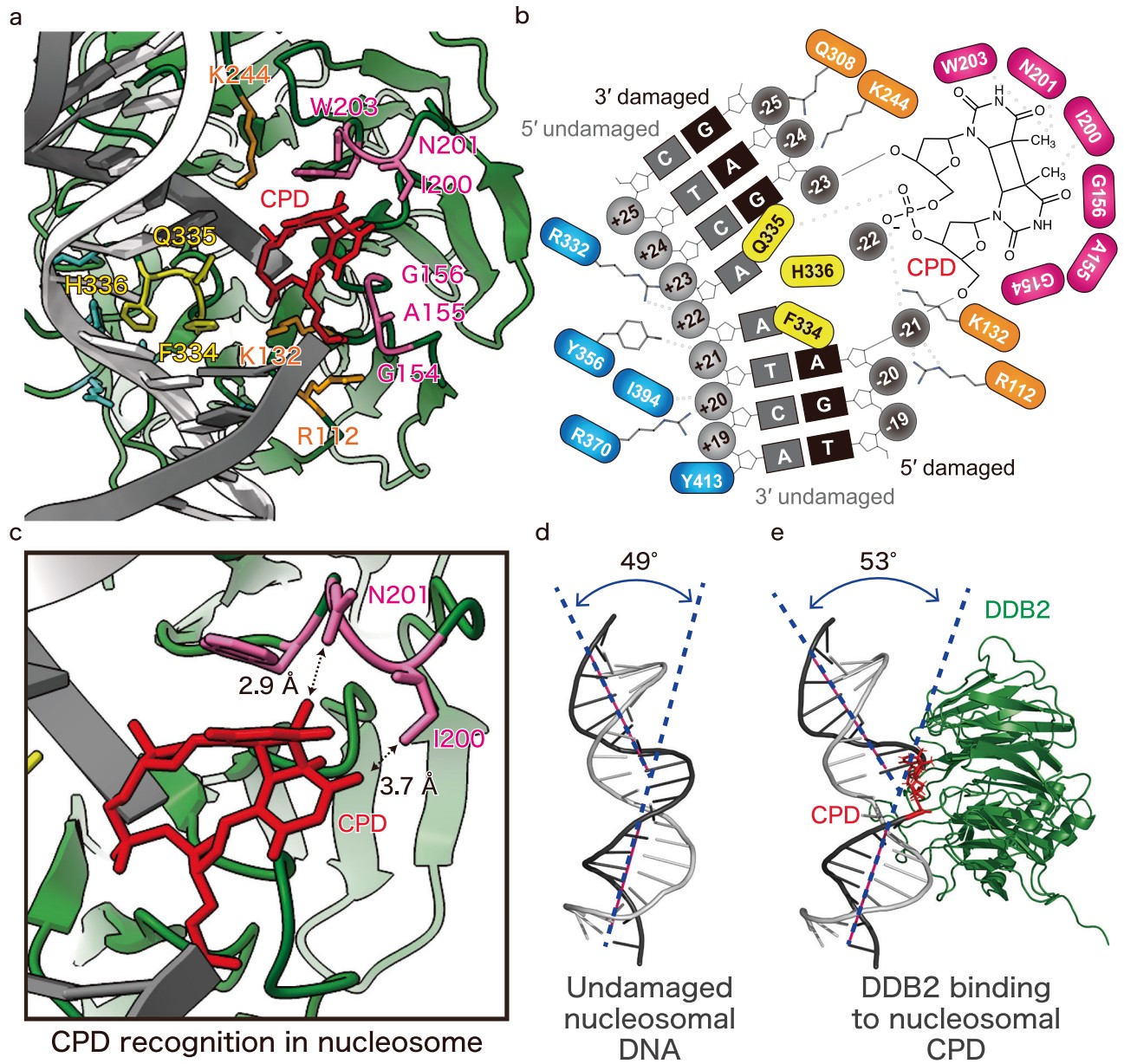

**Fig. 4 | Binding structure of UV-DDB to CPD in the nucleosome. a** Residues contacting the CPD are indicated, and the interacting surface of DDB2 with the CPD (pink), the backbone phosphate of the damaged strand (orange), and the orphan bases of the undamaged strand (yellow) are highlighted. **b** Schematic representation of DDB2 binding to CPD in nucleosomes. **c** Close-up view of contacting residues of DDB2. Distances between CPD and DDB2 residues were measured by the Pymol software. Bending angle of DNA in the nucleosome at the SHL ± 2 position in the absence (**d**) or presence (**e**) of UV-DDB. DNA in the nucleosome without UV-DDB was obtained from the previously determined structural model (PDB ID: 3AFA). Bending angles were measured by generating helical axes of DNA with the 3DNA software[53].

possibly facilitated by nucleosome remodeling activities and/or its own DNA register shifting activity[26].

It should be noted that, in our ChIP-CryoEM analysis, we could not clearly identify the type of DNA lesion because of the limited resolution. The presence of the CPD lesion in the UV-DDB-nucleosome complex may be verified by further immunoprecipitation with anti-CPD antibodies. Further studies will be awaited.

Our pull-down assay in the presence of ATPγS suggested that the damage recognition by UV-DDB in chromatin may depend on ATP hydrolysis. This implies that the ATP-dependent nucleosome remodeling activity may be required for the lesion relocation toward the exposed regions in the nucleosomes, such as the SHL ± 2, ±3, and ±6 positions. Then, UV-DDB may bind to the DNA lesion exposed on the nucleosome surface in cells. UV-DDB reportedly interacts with the

INO80 complex, a chromatin remodeler that functions in nucleosome remodeling and/or removal around DNA lesions[35]. This nucleosome remodeling activity may be important in the DNA-lesion relocation. Further study is required to understand how UV-DDB effectively recognizes the UV lesion by the lesion relocation mechanism in chromatin.

## Methods

### Cell culture and cell line establishment

The Spodoptera frugiperda Sf9 (Expression Systems, catalog# 94-001F) and Trichoplusia ni High Five (Expression Systems, catalog# 94-002F) cell lines were respectively cultured in Sf-900III (Themo Fisher) and ESF 921 (Expression Systems) media containing 1 × Penicillin-Streptomycin (Nacalai Tesque), in suspension at 27 °C.

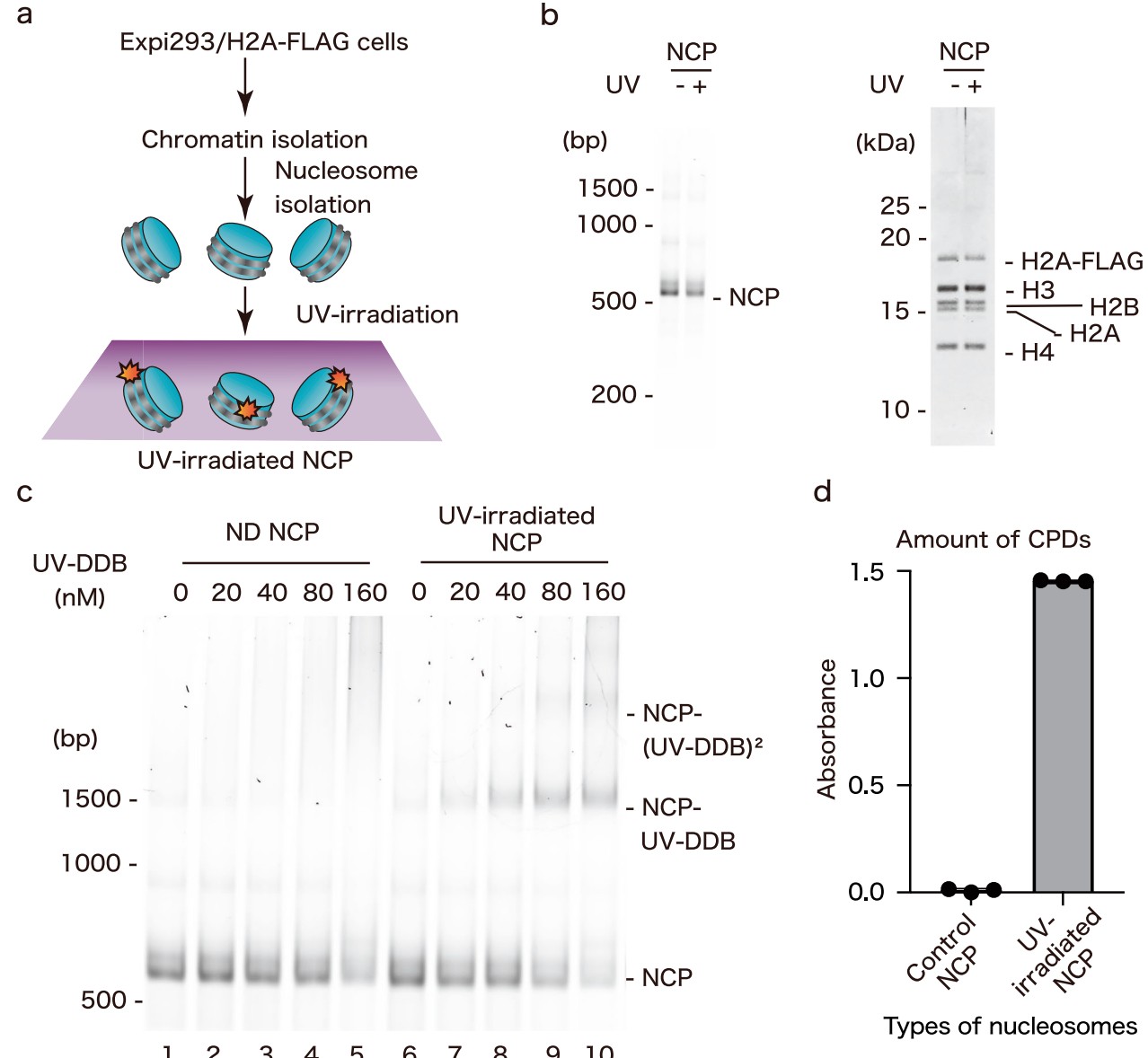

**Fig. 5 | DNA damage recognition by DDB2 in the native nucleosome after UV irradiation. a** Workflow of native nucleosome isolation from expi293 cells and production of native UV-irradiated nucleosomes. **b** Isolated nucleosomes from expi293 cells with or without UV irradiation (500 J/m²) were subjected to Native-PAGE or SDS-PAGE, and then stained with SYBR Gold (left panel) or Oriole (right panel). **c** Gel shift assay, performed using 40 nM of nucleosomes and UV-DDB (0–160 nM). All reactions were subjected to 6% Native-PAGE and DNA was stained by SYBR Gold. **d** Amounts of CPDs in nucleosomes with or without UV irradiation (500 J/m²) were measured by ELISA, using an anti-CPD antibody. The Y-axis represents the absorbance at 490 nm, normalized to the minimum value across all measurements, which was set to zero. The graph was generated using Prism 10 software, and all data are based on three technical replicates and are presented as mean ± s.d.

The U2OS adherent cell line was cultured in RPMI1640 medium (Nacalai Tesque) containing 10% FBS and 1 × Penicillin-Streptomycin, at 37 °C in a 5% CO₂ atmosphere. The U2OS/DDB2 KO cell line was established by K.S. using the CRISPR/Cas9 knockout system[28]. The cDNA encoding N-terminally FLAG-tagged human DDB2 was sub-cloned into the pAAVS1-DNR vector with the AAVS1 Transgene Knock-in vector kit (ORIGENE). Upon co-transfection with pCas-Guide-AAVS1, the U2OS/DDB2 KO/FLAG-DDB2 cell line was selected by 1 µg/ml puromycin. HEK293F (Thermo Fisher, catalog# R79007) expressing FLAG-DDB2 cell line was also established as the same methods using the AAVS1 Transgene Knock-in vector kit (ORIGENE). The Expi293 (Thermo Fisher, catalog# A14635) suspension cell line was purchased from Thermo Fisher and cultured in Expi293 Expression Medium

(Thermo Fisher) containing 0.5 × Penicillin-Streptomycin, at 37 °C in an 8% CO₂ atmosphere. Expi293/H2A-FLAG was established with the same methods described above, by subcloning the cDNA encoding C-terminally FLAG-tagged human H2A into the pAAVS1-DNR vector. The U2OS cell line was authenticated by short tandem repeat (STR) profiling at Promega. The HEK293F and Expi293 cell lines were purchased from Thermo Fisher as pre-validated cell lines.

**Isolation of cellular UV-DDB complex for ChIP-CryoEM**
Cells at 100% confluency (>1 × 10⁸ cells in total) in twenty 150 mm dishes were irradiated with 20 J/m² doses of UV-C under GL-15 germicidal lamps (ActiveRay Scientific) with a 254-nm peak, and then, incubated for 15 min at 37 °C. For the experiments with the ATP analogue

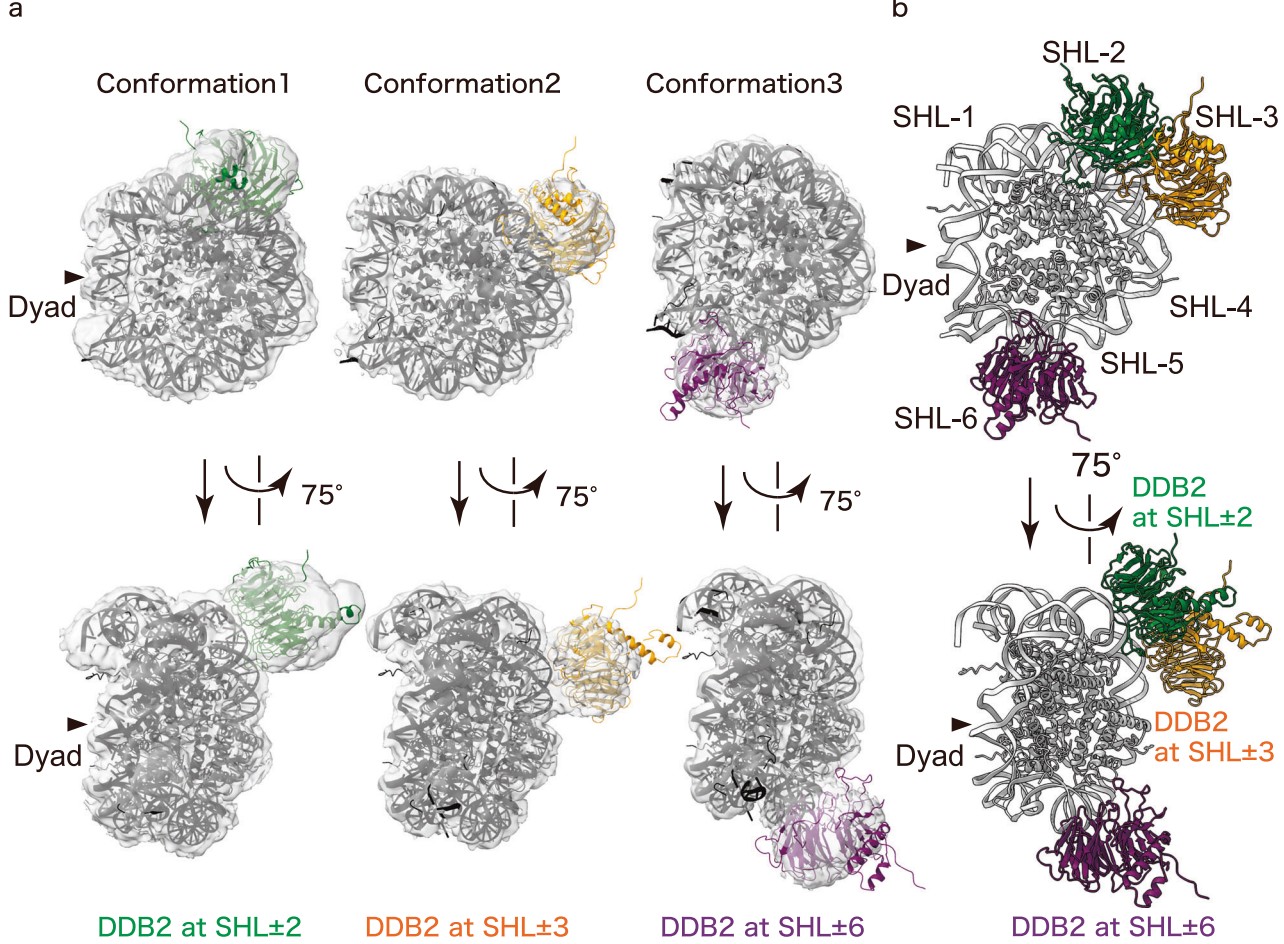

**Fig. 6 | DNA damage recognition of DDB2 at the SHL ± 2, ± 3, and ± 6 positions.**
**a** Cryo-EM maps of three conformations of UV-DDB bound to damaged nucleosomes. 3D structures of the nucleosome (dark gray) and DDB2 (green, orange or purple)(PDB ID 9J8W) were superimposed on cryo-EM maps with masked out DDB1.

Damage locations and DDB2 binding sites of three conformations were predicted as the 22-23 base-pair (SHL ± 2)(green), the 32-33 base-pair (SHL ± 3)(orange), and the 63-64 base-pair (SHL ± 6)(purple). **b** Structural comparison in the merged model of the three conformations in the nucleosome.

treatment, cells were incubated for 8 h in presence of 20 µM Adenosine 5′-O-(3-thiotriphosphate) (ATPγS; Roche) before UV-C irradiation. After two washes with PBS, the cells were treated with 2.5 mg/ml of Trypsin-EDTA for 5 min and then collected with medium. Cells were fixed with 0.75% formaldehyde by an incubation for 10 min at RT, and then the formaldehyde was quenched with a glycine solution (125 µM final concentration). After two washes with PBS, the cells were lysed with CSK buffer (10 mM Pipes-NaOH (pH 6.8), 3 mM $MgCl_2$, 1 mM EGTA, 33.3% sucrose, 0.1% Triton X-100) containing 0.3 M NaCl and protease inhibitor cocktail (SIGMA). After 1 h incubation, the lysates were centrifuged for 10 min at $20,000 \times g$ to separate the soluble and insoluble fractions. The pellets in the insoluble fraction were washed twice with GF buffer, and then the washed pellets were resuspended by an ultra-sonic homogenizer VP050 (TAITEC) for 3 s in GF buffer containing 1 mM $CaCl_2$ and protease inhibitor cocktail. To cleave the chromatin in the insoluble fraction, 1 unit of Micrococcal Nuclease (New England Biolabs) was added per $1 \times 10^3$ cells, and the fraction was incubated for 20 min at 37 °C. After stopping the treatment by adding 1 µl of 0.5 M EDTA per $1 \times 10^6$ cells, the fraction was centrifuged for 10 min at $20,000 \times g$ to obtain the supernatant. The resultant supernatant was loaded onto an affinity column packed with FLAG affinity M2 agarose beads (SIGMA). After washing with GF buffer, the bound proteins were eluted with GF buffer containing 100 µg/ml FLAG-peptide. The eluate was concentrated, and the FLAG-peptide was removed by two rounds of dilution with GF buffer and concentration.

## Immunostaining, immunoblotting and antibodies

The following primary antibodies were used in this study with the indicated dilutions: Anti-DDB2 antibody AF3297 (R&D Systems, catalog# AF3297; 1:500 for immunostaining, 1:2000 for immunoblotting), anti-beta-actin MAB8929 (R&D Systems, catalog# MAB8929, 1:10000), anti-histone H3 D1H2 (Cell Signaling Technology, catalog# 4499, 1:2000), and anti-CPD TDM-2 (Cosmo Bio, catalog# NM-DND-001, 1:1000). Secondary antibodies were used at the following dilutions: Anti-Rabbit IgG (SIGMA, catalog# A9919) and Anti-Goat IgG (SIGMA, catalog# A4062) at 1:10,000; anti-Mouse IgG Alexa Fluor 594 (SIGMA, catalog# A-11032) and anti-Rabbit IgG Alexa Fluor 488 (SIGMA, catalog# A-11034) at 1:500; and anti-Mouse IgG antibody (Rockland, catalog# 710-1602) at 1:2000. For immunofluorescence staining, cells were irradiated with 100 J/m² doses of UV-C by an LED lamp (Altec) with a 265-nm peak through 5 µm pore size hydrophilic polycarbonate membrane filter (Merck) and incubated for 10 min at 37 °C. Cells were fixed with 4% paraformaldehyde for 10 min at RT and permeabilized by 0.5% Triton X-100 for 10 min at RT. Anti-DDB2 and anti-CPD antibodies were applied and incubated for 1 h at 37 °C. After extensive wash, fluorescently labeled secondary antibodies were added and incubated for 1 h at 37 °C. Following further extensive washing, Antifade Mounting Medium with DAPI (Vector Laboratories) were applied, and cells were visualized using a DeltaVision Elite microscope (GE Healthcare) and pseudocoloring of fluorescence images was performed using Fiji software. For immunoblotting, cells were irradiated with 20 J/m²

UV-C (254 nm peak) using GL-15 germicidal lamps (ActiveRay Scientific). After incubation for 30 min, 1 h, or 3 h, cells were lysed in CSK buffer supplemented with 0.3 M NaCl and protease inhibitors. Lysates were fractionated into soluble and insoluble fractions by centrifugation for 10 min at 20,000 × $g$. The resulting pellet was washed twice with CSK buffer, resuspended, and solubilized by sonication using an ultrasonic homogenizer VP050 (TAITEC) for 10 s to obtain the insoluble fraction. Each fraction was separated by SDS-PAGE and transferred to PVDF membranes using the iBlot 2 Gel Transfer Device (Thermo Fisher) according to the manufacturer's P0 program. Membranes were blocked with 10 ml blocking buffer (50 mM Tris-HCl (pH 7.6), 150 mM NaCl, 0.1% Tween-20, 0.1% iBlock (Thermo Fisher)) for 30 min at RT. Primary antibodies (anti-DDB2 AF3297, anti-β-actin MAB8929, anti-histone H3 D1H2) were diluted to the indicated concentrations in 10 ml of the same blocking buffer and incubated with the membranes overnight at 4 °C. After two brief rinses with TBS-T, membranes were washed four times for 10 min each with 10 ml TBS-T. Secondary antibodies diluted in 10 ml blocking buffer were applied and incubated for 1 h RT. Membranes were then washed again four times for 10 min each with 10 mL TBS-T, incubated with CDP-Star substrate (Thermo Fisher) for 5 min at RT, and imaged using an Amersham Imager 680 (Cytiva).

### Enzyme-linked immunosorbent assay (ELISA)

Nucleosome core particles (NCPs) were digested with RNase (NIPPON GENE) and Proteinase K (Roche), and DNA was purified using the QIAquick PCR Purification Kit (QIAGEN). Ten nanograms of heat-denatured DNA were applied to protamine sulfate-coated 96-well ELISA plates (Cosmo Bio, catalog# NM-MA-P003) and air-dried. Wells were washed three times with PBS containing 0.05% Tween-20 (PBS-T), blocked with 2% FBS in PBS at 37 °C for 30 min, and washed again with PBS-T. Samples were incubated sequentially at 37 °C for 30 min each with anti-CPD antibody TDM-2 (Cosmo Bio), anti-mouse IgG secondary antibody (Rockland), and HRP-conjugated streptavidin (Thermo Fisher), with PBS-T washes between steps. Color development was achieved by incubation with citrate–phosphate buffer (24.3 mM citric acid monohydrate, 51.4 mM $Na_2HPO_4$, pH 5.0) for 5 min at RT, followed by addition of o-phenylenediamine (3.7 mM; Fujifilm) and 70 ppm $H_2O_2$ in the same buffer for 5–30 min at 37 °C. Reactions were stopped with 2 M $H_2SO_4$ (final one-third of the reaction volume), and absorbance was measured at 490 nm using a ClarioStar microplate reader (BMG Labtech)[36].

### DNA preparation for nucleosomes

For the preparation of α-satellite DNA sequence, following oligonucleotides were prepared.

Oligo1: ATCAATATCCACCTGCAGATTCTACCAAAAGTGTATTTGGAAACTGCTCCATCAAAAGGCATGTTCAGCTG

Oligo2 (**XX = CPD**):
Phosphate-AACCAGCTGAACATGCCTTTTGA**XX**GAGCAGTTTCCAAATACACTTTTGGTAGAATCTGCAGGTGGATATTGAT

Oligo3:
Phosphate-GTTCAGCTGAACATGCCTTTTGATGGAGCAGTTTCCAAATACACTTTTGGTAGAATCTGCAGGTGGATATTGAT

Oligo4:
ATCAATATCCACCTGCAGATTCTACCAAAAGTGTATTTGGAAACTGCTCCATCAAAAGGCATGTTCAGCTG

While Oligos 1, 3, and 4 were purchased from FASMAC, the oligonucleotide containing cyclobutane pyrimidine dimers (CPD)(Oligo 2) was synthesized on an Applied Biosystem 3400 DNA synthesizer, using the CPD building blocks (Glen Research, catalog#11-1330-90)[37], a chemical phosphorylation reagent (Glen Research, catalog#10-1900-90), and deoxyribonucleoside phosphoramidite building blocks (Glen Research, catalog#10-1000-05, #10-1029-05, #10-1015-05, and #10-1030-05), which

were assembled on controlled-pore glass supports (Glen Research, catalog#20-2031-10). The synthesized oligonucleotides were cleaved and deprotected according to the manufacturer's instructions, and the products were analyzed and purified by HPLC. A Waters μBondasphere C18 5 μm 300 Å column (7.8 mm × 300 mm) was used for purification with a linear gradient of acetonitrile in 0.1 M triethylammonium acetate (pH 7.0). These oligonucleotides were annealed in the combinations of oligos 1 and 3, and oligos 2 and 4, at a 1:1 ratio in annealing buffer (10 mM Tris-HCl pH 8.0, 10 mM NaCl, and 1 mM EDTA). Annealing was performed by heating at 95 °C for 5 min, followed by gradual cooling to 24 °C over 4 h. The resulting duplexes were then ligated at RT for 1 h using T4 DNA ligase (Takara) in 1× ligation buffer (Takara) supplemented with 0.5 mM ATP (Roche). The ligated DNA was further purified by native polyacrylamide gel electrophoresis, using a Prep Cell apparatus (Bio-Rad) in TE buffer (10 mM Tris-HCl (pH 7.5) and 100 μM EDTA). DNA sequences used in this study and CPD locations are indicated in Fig. 3a.

### Protein reconstitution and purification

Human histone proteins were purified from *E. coli*. Equimolar amounts of lyophilized histones were mixed in Denaturing buffer (20 mM Tris-HCl (pH 7.5), containing 7 M guanidine hydrochloride and 20 mM 2-mercaptoethanol). After dialysis against Refolding buffer (2 M NaCl, 10 mM Tris-HCl, pH 7.5, 1 mM EDTA, 2 mM 2-mercaptoethanol), histone complexes were purified by size exclusion column chromatography (HiLoad 16/600 Superdex 200; Cytiva). Fractions containing histone octamers were collected and concentrated. The nucleosome was reconstituted by salt dialysis and purified using 6% non-denaturing polyacrylamide gel electrophoresis, using a Prep Cell apparatus (Bio-Rad). Human strep-tagged DDB1 and strep-tagged DDB2 were subcloned into pAC8-derived vectors (pAC8/strep-DDB1, pAC8/strep-DDB2)[38]. After co-transfection of pAC8 and viral DNA to Spodoptera frugiperda Sf9 cells, the cells were incubated for 5 days. The supernatant (P0 virus) of the medium, which contains the baculovirus expressing strep-DDB1 or strep-DDB2, was collected. Further amplification of the viruses by infecting Sf9 cells for 3 days was repeated until the P3 virus was obtained. DDB1 and DDB2 were co-expressed in 4 L cultures of Trichoplusia ni High Five cells, using the Bac-to-Bac system (Thermo Fisher). Cells were maintained at 27 °C and collected three days post-infection. The pellets were resuspended in lysis buffer (50 mM Tris-HCl pH 8.0, 500 mM NaCl, protease inhibitor cocktail (SIGMA), and 250 μM TCEP) and disrupted by sonication. Following centrifugation, the clarified supernatant was subjected to Streptactin affinity chromatography (IBA) to isolate the UV-DDB complex via the N-terminal Strep-tags on both DDB1 and DDB2. The eluate was subsequently purified using Mono Q ion-exchange chromatography (Cytiva), followed by size-exclusion chromatography (HiLoad 16/600 Superdex 200; Cytiva) in GF buffer (50 mM HEPES pH 7.4, 150 mM NaCl, 250 μM TCEP). The final purified protein was concentrated and stored at −80 °C.

### Gel electrophoretic mobility shift assays (EMSAs)

Nucleosomes (40 nM) were mixed with UV-DDB (0, 10, 20, 40, 80, 160 nM) in Binding buffer (20 mM Tris-HCl, pH 7.5, 150 mM NaCl, 3.4 mM $MgCl_2$, 1.4 mM EDTA, 0.014% Triton X-100, 0.1 mg/ml BSA, and 1 mM DTT) and incubated on ice for 30 min. The mixtures were fractionated on a 6% non-denaturing polyacrylamide gel in 0.5 × TGE buffer (12.5 mM Tris base, 96 mM glycine, and 0.5 mM EDTA). The gel was stained with SYBR Gold (Thermo Fisher) and visualized with a Typhoon biomolecular imaging analyzer (Cytiva).

### Induction of UV lesions to purified nucleosomes

Droplets of purified native nucleosomes (10–15 μl) were spotted in 35 mm glass-bottom dishes (Matsunami) and irradiated with 500 J/m² doses of UV-C by an LED lamp (Aitec) with a 265-nm peak. The droplets were then collected and combined in a microtube.

## Protein identification by LC-MS/MS analysis

Elute fractions were prepared in the same method described in Fig. 1b, c. HEK293F expressing FLAG-DDB2 cells was used in this experiment. Eluted proteins were separated by SDS-PAGE and stained by Oriole Fluorescent (Bio-rad). Target proteins in the excised gels were in-gel digested with trypsin and identified by nano LC-MS/MS analysis according to a modified procedure of references [39–41]. By shotgun method, elute fraction was digested with trypsin and identified by nano LC-MS/MS analysis. LC-MS/MS analysis was conducted by Orbitrap Fusion mass spectrometer equipped with Ultimate3000 nano LC containing an autosampler (Thermo Fisher). The nano LC gradient was delivered at 300 nl/min and consisted of a linear gradient from 2 to 40% acetonitrile in 60 min. Most 10 intense precursor ions were selected in MS1 scans and sequentially fragmented by higher-energy collisional dissociation in MS2 scans. The MS/MS data were searched against the *Homo sapiens* protein sequence database from UniProt using Proteome Discoverer 2.1 (Thermo Fisher) for protein identification. Trypsin cleavage specificity was restricted to lysine residues only, with a minimum peptide length of 6 amino acids and up to two missed cleavage sites allowed. Spectra were searched with a precursor mass tolerance of 10 ppm and a fragment mass tolerance of 0.6 Da. The target false discovery rate (FDR) was set to 0.01 (strict) and 0.05 (relaxed) for peptide-spectrum matches (PSMs) and peptides, and to 0.01 (strict) and 0.05 (relaxed) for proteins. Carbamidomethylation of cysteine was specified as a fixed modification, whereas oxidation of methionine and phosphorylation of serine, threonine, and tyrosine were set as variable modifications.

## Cryo-EM sample preparation

Graphene-coated EM grids were prepared following a modified PMMA-assisted transfer protocol. Briefly, graphene films supported on copper foil were coated with a thin layer of PMMA (poly(methyl methacrylate), average MW 15,000; 1% w/v in anisole) and baked on a hot plate at 100 °C for 15 min. The PMMA/graphene/copper samples were then plasma cleaned using a SOLARUS II plasma cleaner (Gatan) under default $O_2/H_2$ conditions for 2 min. Copper foils were subsequently removed by floating the samples on 1 M ammonium persulfate (APS) solution at RT with repeated exchanges of fresh solution until complete dissolution of the copper substrate. The PMMA/graphene films were then washed several times with MilliQ water before being transferred onto holey gold EM grids (R1.2/1.3, 200 mesh, Au; Quantifoil Micro Tools GmbH). After drying on filter paper, grids were heated on a hot plate at 130 °C for 20 min. To remove the PMMA support, grids were sequentially immersed in acetone, dichloromethane, and 2-propanol at RT, with gentle agitation and incubation steps to ensure complete polymer removal. Finally, grids were dried on filter paper and annealed on a hot plate at 100 °C for 20 min. Grids were stored under ambient conditions until use. Only grids with continuous graphene coverage were selected for subsequent cryo-EM experiments[42]. Glow discharging was performed with ozone generated by a UV253MINI cleaner for 10 min (Filgen). After 8 s blotting, the grid was vitrified using a Vitrobot Mark IV (Thermo Fisher) at 4 °C and 100% humidity. For the structure determination of recombinant UV-DDB bound to NCP^CPD or UV-damaged cellular NCP, UV-DDB was mixed with NCPs, in a 1:2 molar ratio for NCP^CPD or a 1:1 molar ratio for UV-damaged cellular NCP, in GF buffer (50 mM HEPES, pH 7.4, 50 mM NaCl, and 250 μM TCEP). After 30 min incubation on ice, 2.5 μl portions of the samples were applied to R 2/1 Quantifoil grids for NCP^CPD, and R 1.2/1.3 Quantifoil grids for other NCPs. Glow discharging was performed with a PIB-10 ION Bombarder (Vacuum Device) for 60 s under soft conditions. After 6 s of blotting, the grid was vitrified using a Vitrobot Mark IV (Thermo Fisher) at 4 °C and 100% humidity.

## Cryo-EM data acquisition and image processing

Micrographs were captured by a Krios G4 transmission electron microscope (Thermo Fisher) with a K3 direct electron detector and a BioQuantum energy filter (Gatan). Approximately 10,000–20,000 movies were captured for each cryo-EM sample by the EPU software (Thermo Fisher). Data acquisition was performed with a 1.06 Å pixel size and −1.0 to −2.5 μm defocus values at 0.25 μm steps. Data collection was performed by 40 frames. RELION 4.0 and RELION 5.0 were used for further data processing[43]. Motion correction and CTF estimation were performed using MotionCor2 and CTFFIND4, respectively[44,45]. NCP-UV-DDB particles were picked by RELION autopick using a Laplacian of Gaussian filter, and particles were extracted with a binning factor of 2. After obtaining 2D classification images, five images were used as a reference and then reference autopicking and extraction were performed. After junk particles were removed by 2D classification, 3D classification was performed. NCP-UV-DDB classes were selected by several rounds of 3D classification with or without masking, and then possible classes were processed with 3D autorefinement. After re-extraction with a binning factor of 1, 3D autorefinement, Bayesian polishing, and CTF estimation were further applied to obtain high-resolution cryo-EM maps. Final particle sets were analyzed by 3D autorefinement, postprocessing, and the DeepEMhancer software[46].

## Model building and refinement

The initial model of NCP-UV-DDB was built based on the structures previously determined (PDB IDs: 6R8Y, 6R8Z, 5B24, 4A0K). Cryo-EM maps were sharpened after processing with the DeepEMhancer or Relion software as a final map. Models of amino acids were modified by real space refinement with the Phenix software, and adjusted by COOT[47,48]. Structural figures were drawn with the Pymol and ChimeraX software[49,50].

## Reporting summary

Further information on research design is available in the Nature Portfolio Reporting Summary linked to this article.

## Data availability

The cryo-EM maps and coordinates of structures, cellular NCP-UV-DDB, NCP-UV-DDB complex containing CPD, UV-DDB bound to native NCP at SHL ±2, ±3, and ±6 have been deposited in the Electron Microscopy Data Bank (EMDB) and Protein Data Bank (PDB), under the accession codes EMD-61242, EMD-61243 (PDB: 9J8W [https://doi.org/10.2210/pdb9J8W/pdb]), EMD-61246, EMD-61247, and EMD-61248, respectively. The cryo-EM and X-ray crystallographic structures used in this study, NCP-6-4PP(−1)-UV-DDB, NCP_THF2(−1)-UV-DDB, NCP-CPD, DDB1-DDB2-CUL4A-RBX1 bound to a 12 bp abasic site containing DNA-duplex, human nucleosome with the α-satellite DNA sequence, hsDDB1-hsDDB2 complex, and human nucleosome with the Widom 601 L DNA sequence are available in the PDB, under the accession codes 6R8Y [https://doi.org/10.2210/pdb6R8Y/pdb], 6R8Z [https://doi.org/10.2210/pdb6R8Z/pdb], 5B24 [https://doi.org/10.2210/pdb5B24/pdb], 4A0K [https://doi.org/10.2210/pdb4A0K/pdb], 3AFA [https://doi.org/10.2210/pdb3AFA/pdb], 3EI4 [https://doi.org/10.2210/pdb3EI4/pdb], 7VZ4 [https://doi.org/10.2210/pdb7VZ4/pdb]. LC-MS/MS analysis data were deposited in JPOST (ID: JPST003722) and PXD (ID: PXD062246)[51,52]. Source data are provided with this paper.

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

## Acknowledgements

We thank Y. Iikura, Y. Takeda, and M. Dacher (University of Tokyo) for their assistance, and N. Horikoshi (University of Tokyo) for helping to build 3D structure models. This work was supported in part by JSPS KAKENHI Grant Numbers JP22K18034 [to S.M.], JP22K06098 [to Y.T.], JP23H05475 [to H.K.], JP24H02328 [to H.K.], and JP24H02319 [to H.K.]; JST PRESTO Grant Number JPMJPR2288 [to S.M.], JST ERATO Grant Number JPMJER1901 [to H.K.], JST CREST Grant Number JPMJCR24T3 [to H.K.], Novartis Research Grants from The NOVARTIS Foundation for the Promotion of Science [to S.M.], Basic Science Research Projects from The Sumitomo Foundation [to S.M.], and Research Support Project for Life Science and Drug Discovery (BINDS) from AMED under Grant Numbers JP24ama121002 [to Y.T.] and JP24ama121009 [to H.K.].

## Author contributions

S.M., Y.T. and H.K. conceived and planned the experiments. S.M. and K.H. established the cell lines used in this study and prepared samples for western blots, immunostaining, ChIP-CryoEM, ELISA, EMSA, and cryo-EM. L.N. performed mass spectrometry analyses. W.X. prepared the native nucleosomes used in this study. H.T. established the ChIP-CryoEM method for the cellular UV-DDB-NCP complex. S.M., Y.T. and M.O. performed cryo-EM microscopy and analysis. J.Y. and S.I. synthesized CPD-containing oligonucleotides. K.S. established the U2OS/DDB2 KO cell line. S.M., Y.T., K.H., W.X. and H.T. performed biochemical and cryo-EM experiments, with guidance from H.K. All authors contributed to writing the manuscript.

## Competing interests

The authors declare no competing interests.
