## [Transparent Peer Review file · Nature Communications]

Structural basis of cyclobutane pyrimidine dimer recognition by UV-DDB in the nucleosome

Corresponding Author: Professor Hitoshi Kurumizaka

Version 0:

Reviewer comments:

Reviewer #1

(Remarks to the Author)

In this timely and elegant study, Matsumoto et al. report the cryo-EM structures of the UV-DDB complex bound to nucleosomes containing UV-induced DNA lesions, with a specific focus on cyclobutane pyrimidine dimers (CPDs). While their earlier work (Nature, 2019) had already provided critical insight into how UV-DDB recognizes more structurally disruptive lesions such as 6-4 photoproducts (6-4PPs) and abasic (AP) sites in the context of the nucleosome, this current manuscript substantially advances our understanding by showing how UV-DDB recognizes CPDs—the most common UV-induced lesion, but one that causes far less distortion to the DNA helix.

A central finding of the current study is that UV-DDB recognizes CPDs efficiently at solvent-facing superhelical locations (SHL) ± 2 within nucleosomes, without requiring register shifting of nucleosomal DNA. This contrasts with previous observations for 6-4PP and AP sites, where access by UV-DDB necessitated nucleosome remodeling or shifting. The authors show, using a combination of high-resolution in vitro cryo-EM (3.4 Å) and a novel in situ approach - ChIP-CryoEM - that UV-DDB can directly bind CPDs in chromatin isolated from human cells. This is a notable technical and conceptual advance, providing the first structural visualization of UV-DDB engaging physiologically relevant DNA lesions in native nucleosomal contexts.

Furthermore, the study demonstrates that UV-DDB is capable of recognizing UV-induced lesions at additional positions in the nucleosome (SHL ± 3 and ± 6), as visualized in cryo-EM structures of UV-DDB bound to UV-irradiated native human nucleosomes. These findings point to an adaptable lesion recognition mechanism, where UV-DDB engages exposed lesions across a range of accessible nucleosomal positions.

The cryo-EM data are of high quality, particularly the 3.4 Å structure of the UV-DDB–nucleosome complex containing a CPD at SHL ± 2 . The structural details - such as specific contacts between DDB2 residues (e.g., I200, N201, W203) and the CPD moiety - are clearly described and provide new insight into the molecular basis of lesion recognition. The observation that lesion binding induces further DNA bending at the CPD site, even in the already bent context of the nucleosome, is mechanistically intriguing.

This study provides significant and timely insight into how UV-DDB detects CPD lesions in the context of chromatin and is a major contribution to the field of DNA damage recognition and repair. The experiments are carefully designed, the methodologies are sophisticated and well executed, and the data are clearly presented. I am therefore happy to recommend publication of this manuscript, after the following minor points are addressed:

1. Resolution limitations of ChIP-CryoEM. The structure obtained from UV-DDB-nucleosome complexes isolated from cells (via ChIP-CryoEM) is reported at 12.3 Å resolution. While this is sufficient to confirm the approximate binding position of UV-DDB at SHL ± 2 , it would be helpful if the authors could more explicitly discuss the limitations of the resolution in the main text.

2. Confirmation that CPDs are the primary lesions in ChIP-CryoEM samples. The authors infer that the nucleosomal lesions visualized in their ChIP-CryoEM maps are CPDs based on UV-C exposure and ELISA detection. While biologically plausible, a more direct confirmation - perhaps using lesion-specific pulldown or immunoprecipitation, or sequencing-based lesion profiling - would further strengthen this claim. I understand that such experiments may be impossible or impractical, and even a brief discussion of how this limitation could be addressed in future work would be useful.

3. Direct structural comparison with previous 6-4PP-bound complexes. The comparison of UV-DDB binding to CPD versus 6-4PP lesions is central to the manuscript's message. While prior work is cited, a more explicit structural juxtaposition of the new CPD-bound structure and the published 6-4PP-bound complex would help the reader better appreciate the mechanistic distinction..

Reviewer #2

(Remarks to the Author)

Review of Matsumoto et al. – ChIP-CryoEM of UV-DDB on UV-Damaged Chromatin

Matsumoto et al. present an experimental approach termed ChIP-CryoEM to visualize endogenous DNA–protein complexes purified in situ from cells. In this study, they apply the method to UV-DDB (DDB1–DDB2) bound to UV-damaged DNA. The key achievement is the successful isolation and visualization of UV-DDB bound to nucleosomes at SHL ± 2 , in excellent agreement with a conformation previously observed using fully recombinant systems. Importantly, the authors demonstrate that in this context, UV-DDB is engaged with a cyclobutane pyrimidine dimer (CPD), providing the first structural view of a physiologically relevant UV lesion isolated directly from cells.

To validate their findings, the authors reconstitute the system in vitro and solve a high-resolution structure, allowing them to detail the molecular interactions involved in lesion recognition. This dual strategy is particularly compelling for UV-DDB as a test case and solidifies these findings.

A major conceptual advance from this work is the demonstration that SHL ± 2 represents a preferred nucleosomal position for lesion readout, addressing a longstanding question in the field of DNA damage recognition in chromatin. The authors further identify UV-DDB bound to lesions at SHL ± 3 and SHL ± 6 , suggesting that a range of positions can be accommodated in vivo.

While the ChIP-CryoEM reconstructions are intrinsically lower in resolution than those from classical in vitro reconstitutions, this approach uniquely captures the physiological state of nucleosome–lesion complexes on native chromatin substrates. I find the technical achievement of developing this workflow highly impressive, and I commend the authors for the depth of their validation efforts. This work represents, to my knowledge, the first in situ structure of DNA recognition on chromatin, marking a significant advance in the field.

From a methodological perspective, this study also offers an exciting proof-of-principle for ChIP-CryoEM, with broader applicability already exemplified by the companion manuscript on Pol II.

Major comment / suggestion:

The conformations of DDB2 observed here likely represent the main positions at which UV lesions are recognized on nucleosomes. A key remaining question—acknowledged in the final paragraph of the manuscript—is whether CPDs accumulate preferentially at these positions, or whether remodeling activity relocates lesions to SHL ± 2 and related positions. One experimental strategy to address this would be to lyse UV-irradiated cells in the presence of a non-hydrolysable ATP analogue. If lesion repositioning is enzymatically driven, it may be stalled or prevented under such conditions. I believe this experiment would be a valuable addition and is likely feasible within a reasonable timeframe.

Minor comments:

- Figure 2: Where is the lesion located? Are the authors confident in assigning the precise base-pair position? If so, a clearer indication on the figure would enhance interpretability.
- Figure 4d: The comparison of DNA bending angles with the X-ray structure is of limited value due to potential artifacts from crystal packing and the substantial DNA unwinding in the current structure. Consider revising or qualifying this comparison.
- Figure 6: DDB1 is not apparent in the figure. Is this due to resolution or conformational heterogeneity or masking? A brief explanation in the figure legend and main text would be helpful.
- Could the authors comment in the main text whether DDB2 where observed on nucleosomes?
- There is a lengthy discussion on XPC and the putative DDB2 handover complex; this is very hypothetical; this reviewer would rather have seen a discussion on why SHL 2/3/6 emerged as major damage sites on nucleosomes, for example.

Summary: This is an important and technically impressive study that provides the first in situ structure of UV-DDB bound to UV-damaged chromatin. The methodological innovation and biological insight are both substantial. I support publication with minor revisions and would strongly encourage the authors to consider the proposed ATP-analogue experiment, which could further elevate the impact of the manuscript.

Reviewer #3

(Remarks to the Author)

Nature Comm.

UV-DDB

Global genome repair of UV-induced cyclobutane pyrimidine dimers is initiated by UV-DDB in the context of chromatin. Previous work by Drs Sugasawa and N. Thoma (Nature 2019), provided the first cryo-EM structure of UV-DDB binding to a nucleosome containing either a 6-4 photoproduct or abasic sites. Surprisingly they found that UV-DDB binding to these lesions can alter the DNA register on an alpha-satellite nucleosome. In the present work, the team by Drs. Sugasawa and Kurumizaka combined cryo-EM with purified UV-DDB and nucleosomes containing a CPD at SLH -2 with ChIP-cryo-EM pulling down nucleosomes with FLAG2-DDB2 or H2A-FLAG and using the eluate for cryo-EM. This latter approach is innovative.

The authors first present a cryo-EM structure with UV-DDB density near SHL-2 at 12 Angstrom resolution such that side chains and bases were not resolved. They then reconstituted UV-DDB on a CPD at position SLH -2.0. This structure was refined to 3.4 angstroms and afforded good resolution of the CPD flipped out of the helix and the FQH motif inserted into the DNA as was shown previously by in the Thoma cryo-EM nucleosome structure and previous X-ray crystallography structures. Two new contacts were discerned with I200 and N201 contacting the CPD. UV-DDB was found to induce a bend angle of 53 degrees at the site of the CPD, 4 degrees more than unbound undamaged DNA. The using the H2A-FLAG pull down from cells exposed to UV, found a complex array of structures, some consistent with two UV-DDB binding sites on the same nucleosome. Overall, this is an important study that nicely extends their previous work to now include a CPD. The authors should consider the following points that could improve the impact of this work.

1. Introduction: Lines 61-62, while both XP and CS are UV-sensitive, only the former patients suffer skin cancer, the sentence needs to be better stated to indicate this fact.
2. Line 68, might want to indicate that XPC is monoubiquitylated and that poly-Ub of UV-DDB leads to its removal from DNA by p97 and subsequent processing by the 26S proteasome, and cite pertinent papers.
3. Line 126 The ChIP-CryoEM approach that is under review should either be uploaded to BioRxiv or a complete description of the method needs to be included in the Materials and methods. Oddly there is no citation on line 126.
4. The mass spectrometry analysis of proteins that come down with UV-DDB-nucleosome complexes is important. Did the authors have any evidence for Ub-DDB2. It interesting that they detected CUL4B, but not CUL4A or RBX. What were the detection limits with their pull down MS approach? Finally, when discussing these data it would be good to cite the work by Guerrero-Santoro et al, Cancer Res. . 2008 Jul 1;68(13):5014-22 who first showed interacts with CUL4A and CUL4B. It is interesting that no electron density for CUL4B was given in any of the structures. Please comment.
5. While stating that binding of UV-DDB to SHL + 2.0 makes sense for the ChIP-Cryo-EM since the resolution of the structure is not sufficient to follow the polarity of the DNA. The CPD-nucleosome structure shows the CPD at position SHL -2.0 (Figure3) and at that resolution the polarity of the DNA should be clear. Thus stating it as + 2.0 is just not correct. It is essential that the authors provide the specific oligonucleotide sequences that were ligated together to allow reconstitution of the nucleosome as the Thoma Nature paper does not give the actual oligos or protocol. Furthermore a gel of the reconstituted nucleosome should be given.
6. Line 160-161, it would be helpful to align the two structures of the 6-4 and the CPD to show the similarities and differences in the extended Figures.
7. Line 163 – it would be good to point out that the K244E variant of DDB2 no longer recognizes CPD, Ropic-Otrin et al Hum. Mol. Genet., 12 (2003), , and later Ghodke et al, PNAS (2014), please cite these references.
8. Bottom of page 7, lines 190-191. While the pulled down nucleosomes had CPD, that does not prove that the binding observed for UV-DDB is not to 6-4 photoproducts. Thus, while these results are intriguing more discussion that binding could be to 6-4 products which depending on the sequence context can be as high as 3:1 or 2:1 for CPD to 6-4.
9. The interpretation that UV-DDB binds to photoproducts facing outward needs to have the caveat that due to the Thoma paper UV-DDB causes a register shift and thus the work shown in Figure 6 with the various positions could be due to the register shift and not that UV-DDB only detects photoproducts facing outward. Perhaps this is what the authors mean when they state on lines 204-205: "The DNA lesion may relocate to the SHL±2 position, which is more accessible for the nucleosome-binding proteins including UV-DDB". And repeated again on lines 222-224. "Therefore, UV-DDB potentially recognizes UV DNA damage in the nucleosome, when the damaged site is located on the accessible surface." In both places in the text the authors must include the plausible explanation that the structures appear to have the lesion on the surface accessible position on the nucleosome due to the register shift concept of Thoma et al and giving proper citation to this paper in both places.

Version 1:

Reviewer comments:

Reviewer #1

(Remarks to the Author)

The authors have thoroughly addressed all points raised. I am therefore happy to strongly recommend publication.

Reviewer #2

(Remarks to the Author)

My concerns have been sufficiently addressed. I congratulate the authors to this fine piece of work.

Reviewer #3

(Remarks to the Author)

The authors have done a great job of responding to the concerns raised by all three reviewers. This is an important and timely study and will have important impact on the field when it is published.

REVIEWER COMMENTS

Reviewer #1 (Remarks to the Author):

In this timely and elegant study, Matsumoto et al. report the cryo-EM structures of the UV-DDB complex bound to nucleosomes containing UV-induced DNA lesions, with a specific focus on cyclobutane pyrimidine dimers (CPDs). While their earlier work (Nature, 2019) had already provided critical insight into how UV-DDB recognizes more structurally disruptive lesions such as 6-4 photoproducts (6-4PPs) and abasic (AP) sites in the context of the nucleosome, this current manuscript substantially advances our understanding by showing how UV-DDB recognizes CPDs—the most common UV-induced lesion, but one that causes far less distortion to the DNA helix.

A central finding of the current study is that UV-DDB recognizes CPDs efficiently at solvent-facing superhelical locations (SHL) ± 2 within nucleosomes, without requiring register shifting of nucleosomal DNA. This contrasts with previous observations for 6-4PP and AP sites, where access by UV-DDB necessitated nucleosome remodeling or shifting. The authors show, using a combination of high-resolution in vitro cryo-EM (3.4 Å) and a novel in situ approach - ChIP-CryoEM - that UV-DDB can directly bind CPDs in chromatin isolated from human cells. This is a notable technical and conceptual advance, providing the first structural visualization of UV-DDB engaging physiologically relevant DNA lesions in native nucleosomal contexts.

Furthermore, the study demonstrates that UV-DDB is capable of recognizing UV-induced lesions at additional positions in the nucleosome (SHL ± 3 and ± 6), as visualized in cryo-EM structures of UV-DDB bound to UV-irradiated native human nucleosomes. These findings point to an adaptable lesion recognition mechanism, where UV-DDB engages exposed lesions across a range of accessible nucleosomal positions.

The cryo-EM data are of high quality, particularly the 3.4 Å structure of the UV-DDB–nucleosome complex containing a CPD at SHL ± 2 . The structural details - such as specific contacts between DDB2 residues (e.g., I200, N201, W203) and the CPD moiety - are clearly described and provide new insight into the molecular basis of lesion recognition. The observation that lesion binding induces further DNA bending at the CPD site, even in the already bent context of the nucleosome, is mechanistically intriguing.

This study provides significant and timely insight into how UV-DDB detects CPD lesions in the context of chromatin and is a major contribution to the field of DNA damage recognition and repair. The experiments are carefully designed, the methodologies are sophisticated and well executed, and the data are clearly presented. I am therefore happy to recommend publication of this manuscript, after the following minor points are addressed:

Reply)

Thank you very much for this favorable comment. We are happy to revise our manuscript according to your suggestions.

1. Resolution limitations of ChIP-CryoEM. The structure obtained from UV-DDB-nucleosome complexes isolated from cells (via ChIP-CryoEM) is reported at 12.3 Å

resolution. While this is sufficient to confirm the approximate binding position of UV-DDB at SHL ± 2 , it would be helpful if the authors could more explicitly discuss the limitations of the resolution in the main text.

Reply)

We described the reason why the resolution of the ChIP-CryoEM structure of the UV-DDB-nucleosome complex was 12 Å more explicitly in the main text (page 6, lines 136-139).

2. Confirmation that CPDs are the primary lesions in ChIP-CryoEM samples. The authors infer that the nucleosomal lesions visualized in their ChIP-CryoEM maps are CPDs based on UV-C exposure and ELISA detection. While biologically plausible, a more direct confirmation - perhaps using lesion-specific pulldown or immunoprecipitation, or sequencing-based lesion profiling - would further strengthen this claim. I understand that such experiments may be impossible or impractical, and even a brief discussion of how this limitation could be addressed in future work would be useful.

Reply)

Thank you very much for this important comment. As this reviewer noticed, CPD verification in ChIP-CryoEM samples is technically difficult. Immunoprecipitation of ChIP-CryoEM samples with an anti-CPD antibody may work to identify the CPD lesion, but doing so will take more than a year. Therefore, we discuss this issue in the revised text (page 10, lines 255-258), and will save this interesting project for future study.

3. Direct structural comparison with previous 6-4PP-bound complexes. The comparison of UV-DDB binding to CPD versus 6-4PP lesions is central to the manuscript's message. While prior work is cited, a more explicit structural juxtaposition of the new CPD-bound structure and the published 6-4PP-bound complex would help the reader better appreciate the mechanistic distinction.

Reply)

Thank you very much. As this reviewer suggested, we added a direct comparison between NCP-UV-DDB with 6-4PP from the Thomä group and our NCP-UV-DDB with CPD in Extended figure 5f,g.

Reviewer #2 (Remarks to the Author):

Review of Matsumoto et al. – ChIP-CryoEM of UV-DDB on UV-Damaged Chromatin

Matsumoto et al. present an experimental approach termed ChIP-CryoEM to visualize endogenous DNA–protein complexes purified in situ from cells. In this study, they apply the method to UV-DDB (DDB1–DDB2) bound to UV-damaged DNA. The key achievement is the successful isolation and visualization of UV-DDB bound to nucleosomes at SHL ± 2 , in excellent agreement with a conformation previously observed using fully recombinant

systems. Importantly, the authors demonstrate that in this context, UV-DDB is engaged with a cyclobutane pyrimidine dimer (CPD), providing the first structural view of a physiologically relevant UV lesion isolated directly from cells.

To validate their findings, the authors reconstitute the system *in vitro* and solve a high-resolution structure, allowing them to detail the molecular interactions involved in lesion recognition. This dual strategy is particularly compelling for UV-DDB as a test case and solidifies these findings.

A major conceptual advance from this work is the demonstration that SHL ± 2 represents a preferred nucleosomal position for lesion readout, addressing a longstanding question in the field of DNA damage recognition in chromatin. The authors further identify UV-DDB bound to lesions at SHL ± 3 and SHL ± 6 , suggesting that a range of positions can be accommodated *in vivo*.

While the ChIP-CryoEM reconstructions are intrinsically lower in resolution than those from classical *in vitro* reconstitutions, this approach uniquely captures the physiological state of nucleosome-lesion complexes on native chromatin substrates. I find the technical achievement of developing this workflow highly impressive, and I commend the authors for the depth of their validation efforts. This work represents, to my knowledge, the first *in situ* structure of DNA recognition on chromatin, marking a significant advance in the field.

From a methodological perspective, this study also offers an exciting proof-of-principle for ChIP-CryoEM, with broader applicability already exemplified by the companion manuscript on Pol II.

Reply)

Thank you very much for these kind and favorable comments. I appreciate it very much.

Major comment / suggestion:

The conformations of DDB2 observed here likely represent the main positions at which UV lesions are recognized on nucleosomes. A key remaining question—acknowledged in the final paragraph of the manuscript—is whether CPDs accumulate preferentially at these positions, or whether remodeling activity relocates lesions to SHL ± 2 and related positions. One experimental strategy to address this would be to lyse UV-irradiated cells in the presence of a non-hydrolysable ATP analogue. If lesion repositioning is enzymatically driven, it may be stalled or prevented under such conditions. I believe this experiment would be a valuable addition and is likely feasible within a reasonable timeframe.

Reply)

Thank you very much for this insightful suggestion. We repeated the same ChIP-CryoEM experiments in the presence of a non-hydrolyzable ATP analogue, ATP γ S. We then found that, unlike the experiments in the presence of ATP (Fig. 1c), the amounts of immunoprecipitated

DDB2 and histones were not enhanced by pulling down with anti-FLAG antibody. Consistently, our ChIP-CryoEM analysis showed that nucleosomes were hardly detected. These results suggested that the ATP hydrolysis-dependent lesion relocation may be a plausible explanation for the predominant detection of DDB2 bound to the SHL ± 2 position in the ChIP-CryoEM analysis. We added these new experimental data in Extended figure 1 and described that the ATP-dependent repositioning of UV lesions may occur to relocate the UV lesion in the nucleosome for damage recognition in chromatin during nucleotide excision repair, in both the Results (pages 6-7, lines 160-162) and Discussion (page 10, lines 259-264) sections.

Minor comments:

- Figure 2: Where is the lesion located? Are the authors confident in assigning the precise base-pair position? If so, a clearer indication on the figure would enhance interpretability.

Reply)

We highlighted the nucleosomal location of the lesion in Figure 2, together with the superhelical locations.

- Figure 4d: The comparison of DNA bending angles with the X-ray structure is of limited value due to potential artifacts from crystal packing and the substantial DNA unwinding in the current structure. Consider revising or qualifying this comparison.

Reply)

Thank you for this suggestion. To ensure that the DNA bending angle of the X-ray structure is not affected by crystal packing effects, the X-ray structure of the nucleosome used in this comparison is superimposed on a high-resolution cryo-EM structure of the nucleosome without DNA lesions. We then confirmed that these two structures fit very well. These new comparison data between the cryo-EM and X-ray crystallographic models are presented in the new Extended figure 5j, and this fact is described in the text (page 23, line 597).

- Figure 6: DDB1 is not apparent in the figure. Is this due to resolution or conformational heterogeneity or masking? A brief explanation in the figure legend and main text would be helpful.

Reply)

DDB1 is highly flexible, so we masked out the DDB1 subunit to get higher resolution. We added this explanation in the figure legend.

- Could the authors comment in the main text whether DDB2 were observed on nucleosomes?

Reply)

In the ChIP-CryoEM analysis, we obtained the UV-DDB-nucleosome complex structure at 12 Å resolution. This is low resolution, but sufficient to identify the DDB2 and DDB1

subunits in the complex. This fact is now described in the main text of the revised manuscript (page 6, lines 139-140).

- There is a lengthy discussion on XPC and the putative DDB2 handover complex; this is very hypothetical; this reviewer would rather have seen a discussion on why SHL 2/3/6 emerged as major damage sites on nucleosomes, for example.

Reply)

Thank you for this constructive suggestion. Accordingly, in the revised manuscript, we removed the discussion about XPC and the putative DDB2 handover, and focused on the significance of DDB2 binding to SHL \pm 2/3/6.

Summary: This is an important and technically impressive study that provides the first in situ structure of UV-DDB bound to UV-damaged chromatin. The methodological innovation and biological insight are both substantial. I support publication with minor revisions and would strongly encourage the authors to consider the proposed ATP-analogue experiment, which could further elevate the impact of the manuscript.

Reply)

Thank you very much. As mentioned above, we performed the pull-down and ChIP-CryoEM experiments in the presence of a non-hydrolyzable ATP analogue, ATP γ S. These results suggested that the ATP hydrolysis-dependent lesion relocation may be a plausible explanation for the predominant detection of DDB2 bound to the SHL \pm 2 position in the ChIP-CryoEM analysis. We added these new experimental data in Extended figure 1 and described that the ATP-dependent repositioning of UV lesions may occur to relocate the UV lesion in the nucleosome for damage recognition in chromatin during nucleotide excision repair, in both the Results (pages 6-7, lines 160-162) and Discussion (page 10, lines 259-264) sections.

Reviewer #3 (Remarks to the Author):

Nature Comm.

UV-DDB

Global genome repair of UV-induced cyclobutane pyrimidine dimers is initiated by UV-DDB in the context of chromatin. Previous work by Drs Sugasawa and N. Thoma (Nature 2019), provided the first cryo-EM structure of UV-DDB binding to a nucleosome containing either a 6-4 photoproduct or abasic sites. Surprisingly they found that UV-DDB binding to these lesions can alter the DNA register on an alpha-satellite nucleosome. In the present work, the team by Drs. Sugasawa and Kurumizaka combined cryo-EM with purified UV-DDB and nucleosomes containing a CPD at SLH -2 with ChIP-cryo-EM pulling down nucleosomes with FLAG2-DDB2 or H2A-FLAG and using the eluate for cryo-EM. This latter approach is innovative.

The authors first present a cryo-EM structure with UV-DDB density near SHL-2 at 12

Angstrom resolution such that side changes and bases were not resolved. They then reconstituted UV-DDB on a CPD at position SLH -2.0. This structure was refined to 3.4 angstroms and afforded good resolution of the CPD flipped out of the helix and the FQH motif inserted into the DNA as was shown previously by in the Thoma cryo-EM nucleosome structure and previous X-ray crystallography structures. Two new contacts were discerned with I200 and N201 contacting the CPD. UV-DDB was found to induce a bend angle of 53 degrees at the site of the CPD, 4 degrees more than unbound undamaged DNA. The using the H2A-FLAG pull down from cells exposed to UV, found a complex array of structures, some consistent with two UV-DDB binding sites on the same nucleosome. Overall, this is an important study that nicely extends their previous work to now include a CPD. The authors should consider the following points that could improve the impact of this work.

1. Introduction: Lines 61-62, while both XP and CS are UV-sensitive, only the former patients suffer skin cancer, the sentence needs to be better stated to indicate this fact.

Reply)

Thank you for pointing out our confusing sentence. We corrected it accordingly (page 3, line 62).

2. Line 68, might want to indicate that XPC is monoubiquitylated and that poly-Ub of UV-DDB leads to its removal from DNA by p97 and subsequent processing by the 26S proteasome, and cite pertinent papers.

Reply)

According to this suggestion, we added a sentence to explain the lesion-handover system from UV-DDB to XPC via polyubiquitination and p97/VCP segregase, and also cited two reference papers in the revised manuscript (page 3, lines 68-71).

3. Line 126 The ChIP-CryoEM approach that is under review should either be uploaded to BioRxiv or a complete description of the method needs to be included in the Materials and methods. Oddly there is no citation on line 126.

Reply)

The manuscript describing the ChIP-CryoEM technique has now been published in Nature Communications (16(1):4724, 2025). In the revised manuscript, we cited it as Ref. 27.

4. The mass spectrometry analysis of proteins that come down with UV-DDB-nucleosome complexes is important. Did the authors have any evidence for Ub-DDB2. It interesting that they detected CUL4B, but not CUL4A or RBX. What were the detection limits with their pull down MS approach? Finally, when discussing these data it would be good to cite the work by Guerrero-Santoro et al, Cancer Res. . 2008 Jul 1;68(13):5014-22 who first showed interacts with CUL4A and CUL4B. It is interesting that no electron density for CUL4B was given in any of the structures. Please comment.

Reply)

Thank you very much for these insightful comments. We performed a shotgun analysis of the eluted fraction used in Extended figure 1b to search for target proteins. We then found peptides of CUL4A and RBX1, but not Ub-DDB2. We surmised that DDB2 may lose its binding affinity by ubiquitination. This may be a plausible reason why Ub-DDB2 was not detected in our MS analysis. In the revised manuscript, we added these new mass spectrometry data in Extended figure 1c and updated our database, as described on page 5, lines 125-130.

We are keenly interested in the CUL4-ubiquitin ligase in our structure. Unfortunately, we could not observe the cryo-EM density of CUL4-RBX1 in our ChIP-CryoEM samples. An X-ray crystallography analysis by Eric, F.S. et al. showed that CUL4-ubiquitin ligase is highly flexible in the UV-DDB-CUL4-RBX1 complex. This may be the reason why we could not detect CUL4 in our ChIP-CryoEM analysis. This fact is discussed in the revised manuscript (page 6, lines 140-142).

5. While stating that binding of UV-DDB to SHL + 2.0 makes sense for the ChIP-Cryo-EM since the resolution of the structure is not sufficient to follow the polarity of the DNA. The CPD-nucleosome structure shows the CPD at position SHL -2.0 (Figure3) and at that resolution the polarity of the DNA should be clear. Thus stating it as + 2.0 is just not correct. It is essential that the authors provide the specific oligonucleotide sequences that were ligated together to allow reconstitution of the nucleosome as the Thoma Nature paper does not give the actual oligos or protocol. Furthermore a gel of the reconstituted nucleosome should be given.

Reply)

Thank you for this suggestion. We agreed with these points, and corrected the position of the reconstituted nucleosome containing CPD from ± 2 to -2. We also included a gel of the reconstituted nucleosome in Extended figure 3e,f and the protocol for ligation of the oligos and their sequences in the Materials and Methods section.

6. Line 160-161, it would be helpful to align the two structures of the 6-4 and the CPD to show the similarities and differences in the extended Figures.

Reply)

According to this suggestion, we added the comparison figure of 6-4PP and CPD in Extended figure 5f,g, and their similarities and differences are described on page 7, lines 182-185.

7. Line 163 – it would be good to point out that the K244E variant of DDB2 no longer recognizes CPD, Rapic-Otrin et al Hum. Mol. Genet., 12 (2003), , and later Ghodke et al, PNAS (2014), please cite these references.

Reply)

According to this suggestion, we added this sentence “The DDB2 K244E mutant is

reportedly unable to bind to the CPD” in the revised manuscript.

8. Bottom of page 7, lines 190-191. While the pulled down nucleosomes had CPD, that does not prove that the binding observed for UV-DDB is not to 6-4 photoproducts. Thus, while these results are intriguing more discussion that binding could be to 6-4 products which depending on the sequence context can be as high as 3:1 or 2:1 for CPD to 6-4.

Reply)

We agree with your point, and now state that "Consistently, the UV-irradiated nucleosomes were confirmed to contain CPDs, although 6-4PP lesions may also have been present".

9. The interpretation that UV-DDB binds to photoproducts facing outward needs to have the caveat that due to the Thoma paper UV-DDB causes a register shift and thus the work shown in Figure 6 with the various positions could be due to the register shift and not that UV-DDB only detects photoproducts facing outward. Perhaps this is what the authors mean when they state on lines 204-205: “The DNA lesion may relocate to the SHL \pm 2 position, which is more accessible for the nucleosome-binding proteins including UV-DDB”. And repeated again on lines 222-224. “Therefore, UV-DDB potentially recognizes UV DNA damage in the nucleosome, when the damaged site is located on the accessible surface.” In both places in the text the authors must include the plausible explanation that the structures appear to have the lesion on the surface accessible position on the nucleosome due to the register shift concept of Thoma et al and giving proper citation to this paper in both places.

Reply)

Thank you very much for this comment. In the manuscript, we added this sentence: “In this process, the DNA register-shifting activity of UV-DDB may also contribute to lesion relocation” for the first caveat, and rewrote the sentence related to the second caveat: “Therefore, UV-DDB may recognize UV-induced DNA damage in the nucleosome when the damaged site is located on an accessible surface, possibly facilitated by nucleosome remodeling activities and/or its own DNA register-shifting activity” with the citation for register shifting published by Thomä’s group.